

# Distribution, habitat suitability, conservation state and natural history of endangered salamander *Bolitoglossa pandi*

Teddy Angarita-Sierra[1,2,3], M. Argenis Bonilla-Gómez[3], David A. Sánchez[1,4], Andres R. Acosta-Galvis[5], Hefzi Medina-Ovalle[3], Anggi Solano-Moreno[3], Simon Ulloa-Rengifo[3], Daniela Guevara-Guevara[3], Juan J. Torres-Ramirez[1,3], Sebastián Curaca-Fierro[3], Diego M. Cabrera-Amaya[1], Jhon A. Infante-Betancour[1], Luisa F. Londoño-Montaño[1], Diana X. Albarán-Montoya[6] and Lesly R. Peña-Baez[6]

[1] YOLUKA ONG, Fundación de investigación en Biodiversidad y Conservación, Bogotá, Colombia
[2] Vicerectoria de investigación, Universidad Manuela Beltrán, Bogotá, Colombia
[3] Grupo de Investigación Biología de Organismos Tropicales (BIOTUN), Departamento de biología, Universidad Nacional de Colombia, Bogotá, Colombia
[4] Instituto Amazónico de Investigaciones Científicas SINCHI, Leticia, Amazonas, Colombia
[5] Colecciones Biológicas IAvH, Subdirección de Investigaciones, Instituto de Investigación de Recursos Biológicos Alexander von Humboldt, Villa de Leyva, Boyacá, Colombia
[6] Fundación Ecotrópico Colombia, Bogotá, Colombia

Corresponding author
Teddy Angarita-Sierra, tgangaritas@unal.edu.co

## ABSTRACT

**Background**. Pandi's mushroom-tongue salamander (*Bolitoglossa pandi*) is one of the threatened amphibians in South America, as well as a flagship species for the Colombian conservation agenda. This species is endemic to the Andean cloud forests of the western slope of the Cordillera Oriental of Colombia, occurring only in the department of Cundinamarca within a narrow elevational range. At night, *B. pandi* can be seen perching on the upper side of leaves at heights ranging from ground level to 2.5 m. During the day, it can be found under leaf litter or cover objects. Few studies have provided relevant information that can help the Colombian government to formulate lines of action for the conservation of this species; consequently, its threat assessments so far have been based on very limited information.
**Methods**. We conducted surveys for salamanders in four municipalities of Cundinamarca, Colombia, using two approaches: visual encounter surveys (Guaduas and Villeta) and the basic sampling protocol for single-species occupancy modeling (Supatá and Venecia). Multivariate analyses were employed to explore the correlation between habitat structure and natural history traits, abundance, and detection/non-detection of *B. pandi*. We evaluated the *B. pandi* activity pattern through kernel density curves for each sampling occasion and explored the variability of salamander abundance during their activity period by performing a nested ANOVA.
**Results**. We report the discovery of two new populations of *B. pandi,* which represent the most northwestern records known. A significant correlation between body length, body mass, and habitat structure was observed. Multivariate analyses indicated that leaf litter depth, mean temperature, percent vegetation cover, and altitude were the habitat variables that together explained 60.3% of the *B. pandi* abundance variability, as well

as the main determinants of its optimal habitat. *Bolitoglossa pandi* exhibits an activity pattern characterized by two main activity peaks, in which niche time-partitioning was observed. Across the surveyed area, we found a healthy, stable, highly dense population of *B. pandi* ($>$1,300 individuals), with seasonal variability between development stages. **Discussion**. Given the high habitat specificity of *B. pandi,* the species is highly vulnerable to local changes. Thus, we recommend that *B. pandi* be retained as Endangered (EN) on the IUCN Red List, based on the IUCN Criterion B, given its restricted extent of occurrence (ca. 2,500 km$^2$) and the ongoing threats from agriculture, cattle ranching, logging, and urban development, which continue to reduce its suitable habitat.

## INTRODUCTION

Colombia is home to more than 850 species of amphibians, being mainly concentrated in the Andean Forest (*Acosta-Galvis, 2020*). Geographic distributions of fauna in this biome are often restricted and endemism is common (*Lynch & Suárez-Mayorga, 2002*). Andean forests are among the most threatened habitats in the country due to the drastic transformation of native vegetation by urban growth, mining, agriculture, and cattle ranching (*Etter et al., 2018*). As a consequence and given that most threatened Colombian amphibian species are concentrated between 1,800–3,600 m a.s.l., habitat loss is the main threat shared by most species.

Among threatened Colombian amphibians, salamanders of the genus *Bolitoglossa* are one of the flagship amphibian groups on the Colombian conservation agenda. This genus is the most diverse and geographically widespread lineage of plethodontid salamanders inhabiting the Western Hemisphere. Currently, *Bolitoglossa* comprises 134 species, 24 of which reside in Colombia across several types of tropical habitats (*Wake, 2017*; *Acosta-Galvis, 2020*; *Frost, 2020*). Many species of *Bolitoglossa* exhibit restricted geographic ranges (e.g., *Bolitoglossa capitana* (*Brame & Wake, 1963*), *B. hypacra* (*Brame & Wake, 1962*), *B. hiemalis* (*Lynch, 2001*), among others). However, the high level of morphological crypsis and an incomplete understanding of the morphological variability among Andean *Bolitoglossa* species make suitable determinations regarding their distributional ranges hard to achieve (*Acosta-Galvis & Gutiérrez-Lamus, 2012*).

The ecology and life history of South American *Bolitoglossa* salamanders are poorly understood compared to their Central and North American congeners; less than 12% of South American *Bolitoglossa* species have been investigated with published information on their diet, reproduction, foraging activities, thermal ecology, demography, ecological interactions, microhabitat use, or habitat preferences (*Houck, 1977*; *Jimenez, 1994*; *Bruce, 1997*; *Salgado-Aráuz, 2005*; *Anderson & Mathis, 2006*; *Cadenas et al., 2009*; *Ortega, Monares-Riaño & Ramírez-Pinilla, 2009*; *Neckel-Oliveira et al., 2011*; *Del Río-García, Serrano-Cardozo & Ramírez-Pinilla, 2014*; *Cruz, Galindo & Bernal, 2016*). Given this dearth

of information, the conservation status of many of these species is Data Deficient (DD) or has been based on limited information.

According to the IUCN red list of threatened species, *Bolitoglossa pandi* (*Brame & Wake, 1963*) is Endangered B1ab (iii), based on its restricted geographical range. It was originally described based on a single specimen (holotype ZSZMH 2858, an adult female with Snout-vent length 50.4 mm, collected in 1913 by Wilhelm Frietsche) from the municipality of Pandi, Cundinamarca (exact locality within the municipality is unknown), in the cloud forests on the western slope of the Cordillera Oriental of Colombia. Subsequently, *Hanken & Wake (1982)* reported a second specimen 75 km north airline from the type locality, in a bromeliad inside primary cloud forest, near the municipality of Albán (Cundinamarca), at 2,400 m a.s.l.

*Acosta-Galvis & Rueda-Almonacid (2004)* reported a third specimen (an adult female with snout-vent length (SVL) 44.7 mm, collected by Franz Kaston, ICN 45500) near the municipality of Pandi and described the associated habitat as relict wet areas covered by lush trees and shrubs. *Acosta-Galvis & Gutiérrez-Lamus (2012)* included a new record for the Supatá region (adult male with SVL 37.63 mm, MUJ 7921) representing the northernmost record of the species. Hence, the known distribution of *B. pandi* includes four localities from the sub-Andean forests between 1,300–2,400 m a.s.l., throughout the western slopes of the Cordillera Oriental. Since its first extinction risk assessment was performed 14 years ago, few studies have provided relevant information for the formulation of strategies or an action plan for its conservation (*Del Río-García, Serrano-Cardozo & Ramírez-Pinilla, 2014*).

The aims in this study are: (1) to describe the geographic range of *B. pandi* along the western slopes of the Cordillera Oriental; (2) to explore the relationship between habitat structure and natural history traits of this species; (3) to describe its activity pattern and population stage-structure; (4) to expand the knowledge of the variability of certain morphological characters and life-history traits of this poorly known species; (5) and to provide a conservation status reassessment of *B. pandi*.

## MATERIALS & METHODS

### Ethics statement

Sex was not determined on living salamanders due to the high risk of injury to the animal. Fieldwork was done under the scientific research non-commercial purpose permit of collection of wild specimens of biological diversity issued by the National University of Colombia (Research Project 38615), and the Colombian National Environmental Licensing Authority (ANLA) by resolution No. 0255 of 14 March 2014. This study was conducted following the Colombian animal welfare law and the collection of wild specimens of the biological diversity acts (Ley 1774, 2016; Decreto 1376, 2013), as well as considering the Universal Declaration on Animal Welfare (UDAW) endorsed by Colombia in 2007.

### Study area

We searched for salamanders at nine localities in four municipalities of Cundinamarca located on the western slope of the Cordillera Oriental of Colombia: Guaduas, Supatá,

Venecia and Villeta (Fig. 1). Searches at each locality were carried out within an elevational gradient ranging from 1,638 to 2,315 m a.s.l (Table 1). The sampled area includes sub-Andean and Andean forests, as well as areas transformed by urban growth, agriculture, and cattle ranching. The sampling area is characterized by a bimodal climate [high dry season (from mid-December to mid-March); high rainy season (from mid-March through June); low dry season (from July to mid-September), low rainy season (from mid-September to mid-December)]. We monitored the environmental temperature (ET) and relative humidity (RH) at sampling sites using Ebro® thermo-hygrometers (model EBI 20-TH1).

## Sampling and data collection

We conducted surveys for salamanders using two approaches. First, during the rainy season in April–May 2013, we performed visual encounter surveys (VES; *Crump & Scott, 1994*) in three localities associated with cloud forests throughout an altitudinal gradient (1,648–2,002 m a.s.l) in the municipalities of Guaduas and Villeta (Table 1). Two researchers surveyed day and night for five days, for a total of 100 h of sampling effort.

Second, we employed the basic sampling protocol described by *Mackenzie et al. (2003)* for single-species occupancy modeling in six localities, three in the municipality of Supatá and three in Venecia. We randomly selected a total of 296 plots (5 m × 5 m), which were located throughout an altitudinal gradient at each sampling locality (1,637–2,315 m a.s.l), grouping the following vegetation covers: Andean forest fragments, restored Andean riparian forest, pastures and roadsides (Table 1). During three sampling occasions (September–October 2017, March 2018, and July 2018), each plot was surveyed day and night for five consecutive days by ten researchers, resulting in a total of 2100 h of sampling effort. During each survey, the detection/non-detection of *B. pandi* specimens were recorded. When a salamander was present, we measured its perch height using a measuring tape (±0.1 cm). Once salamanders were caught, we recorded their weight with a Pesola® dynamometer of 50 g (±0.1 g) and took photographs to measure their body size (SVL, snout-vent length (mm); TL, Tail length (mm)) using the software Image–J v. 1.52 (*Bourne, 2010*). All specimens of *B. pandi* were subsequently released near the plot where they were sighted.

Based on 18 specimens collected, euthanized using 2% lidocaine, and fixed in 10% formalin (*Chen & Combs, 2001*), we described the morphological variability of *B. pandi*. We made a small incision in the groin region to identify their sex and sexual maturity through macroscopic observation of the gonads. All the morphological terminology employed follows several contributions (*Brame & Wake, 1962*; *Brame & Wake, 1963*; *Acosta-Galvis & Restrepo, 2001*; *Lynch, 2001*; *Acosta-Galvis & Hoyos, 2006*; *Acosta-Galvis & Gutiérrez-Lamus, 2012*; *Brcko, Hoogmoed & Neckel-Oliveira, 2013*; *Bingham et al., 2018*). All specimens were deposited in the amphibian collection at Instituto de Investigación de Recursos Biológicos Alexander von Humboldt, (IAvH-Am), as well as in the amphibian collection at the Instituto de Ciencias Naturales de la Universidad Nacional de Colombia (ICN).

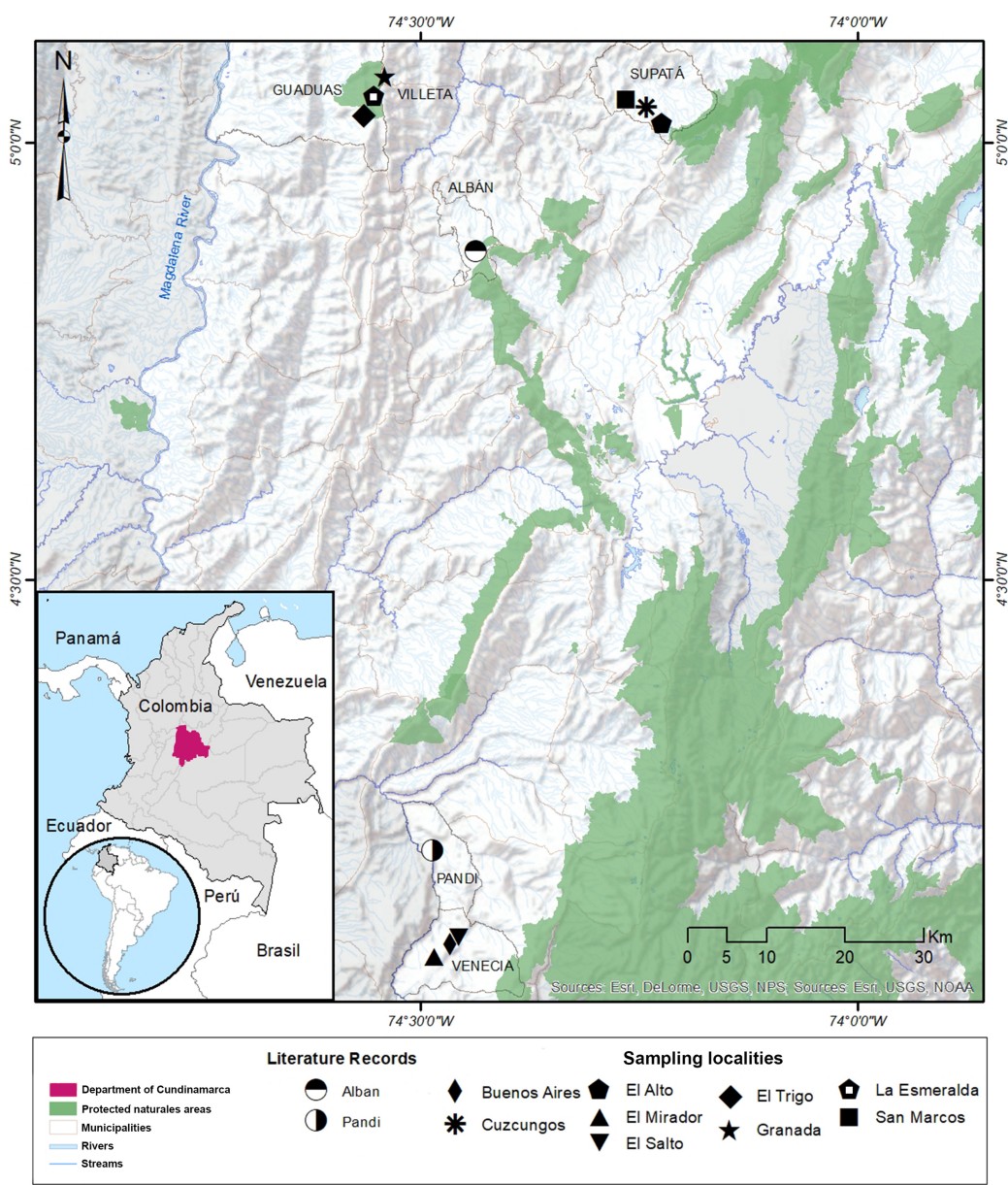

**Figure 1** **Distribution range of the poorly-known salamander _Bolitoglossa pandi_.** Distribution range of the poorly known salamander _Bolitoglossa pandi_. New northwestern records for the known distribution of _Bolitoglossa pandi_ (IAvH-Am 10303-4, IAvH-Am 10305-8). Pentagon with a white square inside: La Esmeralda (municipality of Villeta), black star: Granada (municipality of Villeta), broad diamond: El Trigo (municipality of Guaduas). Sampled localities in the municipality of Supatá. Black square: San Marcos, asterisk: Cuzcungos Natural Reserve, solid pentagon: "El Alto" vereda Monterey. Sampled localities in the municipality of Venecia. Narrow diamond: Buenos Aires; inverted triangle: El salto de la Chorrera; triangle: El Mirador; half-filled circles: Literature records. Background map was retrieved from the Esri open database accessing the following sources: DeLorme, USDS, NPS; USGS, NOAA.

**Table 1  Sampling localities on the western slope of the Cordillera Oriental of Colombia.**

| Municipalities Vereda/locality | Latitude North | Longitude West | Elevation (m) a.s.l | Sampling protocol | *B. pandi* specimens observed | *B. pandi* specimens collected |
|---|---|---|---|---|---|---|
| **Supatá** | | | | | | |
| Vereda Las Lajas/Monterey | From 5°2′18.5″ To 5°1′59.9″ | 74°14′6.6″ 74°14′3.5″ | 2119–2315 | P ($N = 45$) | 1 | 0 |
| Vereda Las Lajas/ Cuzcungos Natural Reserve | From 5°2′27.7″ To 5°2′27.2″ | 74°14′32.2″ 74°14′27.9″ | 1931–2016 | P ($N = 43$) | 1391 | 12 |
| Vereda San Marcos | From 5°2′58.9″ To 5°2′58.3″ | 74°15′53.7″ 74°15′54.9″ | 1743–1773 | P ($N = 28$) | 0 | 0 |
| **Villeta** | | | | | | |
| Vereda La Esmeralda | 5°3′17.9″ | 74°32′49.1″ | 1996 | VES | 10 | 2 |
| **Guaduas** | | | | | | |
| Vereda Granada | 5°4′1.49″ | 74°32′59.6 | 1816 | VES | 16 | 4 |
| El Trigo site | 5°2′9.2″ | 74°33′48.8″ | 1650 | VES | 8 | 0 |
| **Venecia** | | | | | | |
| Vereda Buenos Aires | From 4°4′57.9″ To 4°4′44.1″ | 74°27′57.7″ 74°27′33.3″ | 1809–2128 | P ($N = 90$) | 0 | 0 |
| Vereda El Diamante/El salto de la Chorrera | From 4°5′24.6″ To 4°5′21.0″ | 74°27′25.5″ 74°27′33.2″ | 1637–1735 | P ($N = 36$) | 0 | 0 |
| Vereda El Alto/ Road Venecia- Cabrera | From 4°4′5.2″ To 4°4′11.1″ | 74°29′6.9″ 74°28′54.8″ | 2034–2114 | P ($N = 54$) | 0 | 0 |

**Notes.**

VES,  Visual encounter surveys (*Crump & Scott, 1994*).

P = Plots

N = Number of plots randomly selected

## Habitat structure data collection

We used the Point Intercept Method described by *Elzinga, Salzer & Willoughby (1998)* to estimate the percent vegetation cover. We grouped plants into eight growth forms: graminoids, forbs, palm trees, mosses, lichens, vines, shrubs, and trees. We divided vegetation into layers: 0–0.1 m, 0.1–1 m, 1–1.5 m, 1.5–3 m, 3–5 m, and 5–12 m. We also considered ground characteristics such as leaf litter, bare soil and bare rocks. We estimated the vegetation cover of each of the plots where salamanders were surveyed, employing a set of 15 intercept points distributed in three parallel lines of five points separated by one meter of distance. At each point, we used a sampling bar of 1.5 m to register the contact of the growth forms of each vegetation layer below 1.5 m. This provided us with a 6.67% cover resolution by layer. We assessed the percentage cover of the vegetation layers above 1.5 m (mostly trees) using five intercept points: the corners of the quadrant and the central point of the third line. In this way, we reached a 20% cover resolution for upper layers.

## Statistical analysis

We evaluated the association between habitat structure and the natural history traits of *B. pandi* by multiple correlation analysis, with $P < 0.05$ as the significance level. The following variables were considered: SVL (mm), weight (g), perch height (mm), leaf litter depth (mm), vegetation layers, vegetation life form, percent vegetation cover and elevation

(meters above sea level). Using the habitat variables, we performed a principal component analysis (PCA) to explore which of these variables presented greater variability between plots with regards the detection/non-detection of *B. pandi* and, therefore, which of these could explain the observed differences between plots. The variable suitability for PCA was tested performing a Kaiser–Meyer–Olkin test (KMO > 0.5, $P < 0.05$). Afterward, a quadratic discriminant analysis was performed to determine which of the habitat variables had the greatest discrimination capacity between plots where *B. pandi* was detected/non-detected.

We assessed the variability in salamander abundance observed in the Supatá population through multiple regression analysis. First, we considered the salamander abundance as a dependent variable and the habitat structure variables recorded at each sampling plot as independent variables: leaf litter depth (mm), vegetation layers, vegetation growth forms, percent vegetation cover, elevation (meters above sea level), temperature (°C), and environment relative humidity. All variables were Ln–transformed prior to perform the statistical analysis.

Second, we evaluated assumptions of normality, autocorrelation, and homoscedasticity using Kolmogorov–Smirnov test, Durbin–Watson test and Breusch-Pagan test, respectively. Given that the *p*–value of the Durbin–Watson test can easily be less than 0.05 when sample size is very large, we used the Durbin–Watson statistic test (DW) as an autocorrelation criterion. According to *Durbin & Watson (1950)*, a DW of less than 1 indicates a strong positive autocorrelation, a DW greater than 4 indicates a strong negative autocorrelation, values between 1 and 3 suggest a moderate autocorrelation, and a value close to 2 means that there is no autocorrelation.

Third, we tested for multicollinearity between the variables using the variance inflation factor (VIF) with a threshold of 10. Fourth, we selected the "best" regression model employing the Akaike Information Criterion (AIC; *Akaike, 1973*), considering that models with ΔAIC values of less than two are equally plausible (*White & Burnham, 1999*). Finally, we used the hierarchical partitioning method to evaluate the contribution of all the independent variables of the regression model (*Chevan & Sutherland, 1991*).

## Activity pattern and population stage-structure

We only assessed the activity pattern and the population stage-structure of the *B. pandi* population at the Cuzcungos locality given the high abundance observed (Table 1). We estimated the activity pattern through kernel density curves for each sampling occasion, as well as combining all sampling occasions. We explored the salamander abundance variability during their activity period by performing a nested ANOVA. Hence, the activity period of *B. pandi* was divided into nine time-intervals (from H1 = 18:30–19:30, H2 = 19:31–20:30, H3 = …until H9 = 02:31–03:30), and each salamander sighting was allocated into its respective interval. The sampling occasion was used as the primary factor, and the time intervals as the secondary factor nested in the primary factor. We evaluated assumptions of normality and homogeneity of variances using a Shapiro–Wilk test and Levene test, respectively. Additionally, we analyzed the variability in observed body size over the nine time-intervals through a non-parametric ANOVA using a Kruskal-Wallis test (KW) as a measure of the central tendency of the samples (*Sokal & Rohlf, 1981*).

We used SVL as a descriptive variable of the population stage-structure of *B. pandi*. We compared the variability in population stage-structure among sampling occasions through a Wilcoxon test, with the null hypothesis being that population median stage-structure was the same across all sampling occasions. According to the categories proposed by *Acosta-Galvis & Gutiérrez-Lamus (2012)* and *Del Río-García, Serrano-Cardozo & Ramírez-Pinilla (2014)*, as well as the development stage of the collected salamanders, the population was divided into stage classes as follows: neonates (≤23 mm), juveniles (24–30 mm), and adults (≥30 mm).

All statistical analyses were performed using the software Rwizard 4.3 (*Guisande et al., 2014*) and the following R packages: car (*Fox & Weisberg, 2019*), hier.part (*Nally & Walsh, 2004*), lawstat (*Hui, Gel & Gastwirth, 2008*), nortest (*Gross & Ligges, 2015*), overlap (*Ridout & Linkie, 2009*) stat (*Bolar, 2019*) and usdm (*Naimi et al., 2014*).

# RESULTS

## Geographic distribution

We found a total of 34 *B. pandi* individuals at three new localities, extending the geographical range of the species by 96.5 Km (airline) northwest from the type-locality, and 33.6 km (airline) west from the northernmost locality in the municipality of Supatá (Fig. 1). These new localities belong to the municipalities of Guaduas and Villeta in the department of Cundinamarca. All the salamanders in the new localities were found at night, within the understory of oak groves dominated by ferns. The salamanders were found in different vertical strata ranging from leaf litter, where they remained hidden, to shrubby substrates up to 2.5 m. Additionally, two salamanders were found in ecotonal areas associated with sugar cane crops and rangeland areas for livestock.

## Associations of habitat structure and natural history traits

We observed a significant association between habitat structure and morphological traits. Snout-vent length, tail length, and mass were significantly associated with all the habitat structure variables assessed, but less so with the vegetation layers. SVL, TL, and mass showed a negative correlation with leaf litter and elevation (Table 2). In contrast, SVL and mass showed a positive correlation with perch height and percent vegetation cover (Table 2, Fig. 2).

Similarly, the habitat structure variables were significantly associated with the detection or non-detection of *B. pandi* throughout the sampling plots. The first two components of the PCA accounted for 62.4% of the variability observed. The habitat structure variability was clustered in two groups associated with the detection or non-detection of *B. pandi* (Fig. 3A). These groups were moderately overlapping in multivariate space, but they were differentiated by elevation, leaf litter depth, and percent vegetation cover. The presence of *B. pandi* was positively correlated with highly structured plots and with deep leaf litter (Table 3). The quadratic discriminant analysis confirms that the detection or non-detection of *B. pandi* depends on habitat variables. The cross-validation percentage was 92.6%, indicating that the quadrants in which *B. pandi* was detected can be clearly

**Table 2 Multiple correlation analysis.** The upper diagonal part contains correlation coefficient estimates. The lower diagonal part contains corresponding *p*-values. Bold values denote statistical significance at the $p < 0.05$ level.

| Habitat structure variable | SVL (mm) | TL (mm) | Mass (g) | Perch height (mm) | Growth forms | Percent vegetation cover | Vegetation layers | Leaf litter depth | Elevation (m) a.s.l |
|---|---|---|---|---|---|---|---|---|---|
| SVL (mm) | ***** | 0.890 | 0.901 | 0.316 | 0.216 | 0.287 | 0.006 | −0.306 | −0.155 |
| TL (mm) | **<0.001** | ***** | 0.820 | 0.281 | 0.241 | 0.262 | −0.037 | −0.306 | −0.126 |
| Mass (g) | **<0.001** | **<0.001** | ***** | 0.288 | 0.194 | 0.271 | 0.018 | −0.266 | −0.134 |
| Perch height (mm) | **<0.001** | **<0.001** | **<0.001** | ***** | 0.084 | 0.141 | 0.029 | −0.058 | 0.004 |
| Growth forms | **<0.001** | **<0.001** | **<0.001** | **0.026** | ***** | 0.490 | −0.039 | −0.448 | −0.043 |
| Percent vegetation cover | **<0.001** | **<0.001** | **<0.001** | **<0.001** | **<0.001** | ***** | 0.063 | −0.396 | −0.152 |
| Vegetation layers | 0.883 | 0.322 | 0.627 | 0.445 | 0.300 | 0.096 | ***** | 0.203 | 0.133 |
| Leaf litter depth | **<0.001** | **<0.001** | **<0.001** | 0.125 | **<0.001** | **<0.001** | **<0.001** | ***** | 0.092 |
| Elevation (m) a.s.l | **<0.001** | **0.001** | **<0.001** | 0.906 | 0.260 | **<0.001** | **<0.001** | **0.015** | ***** |

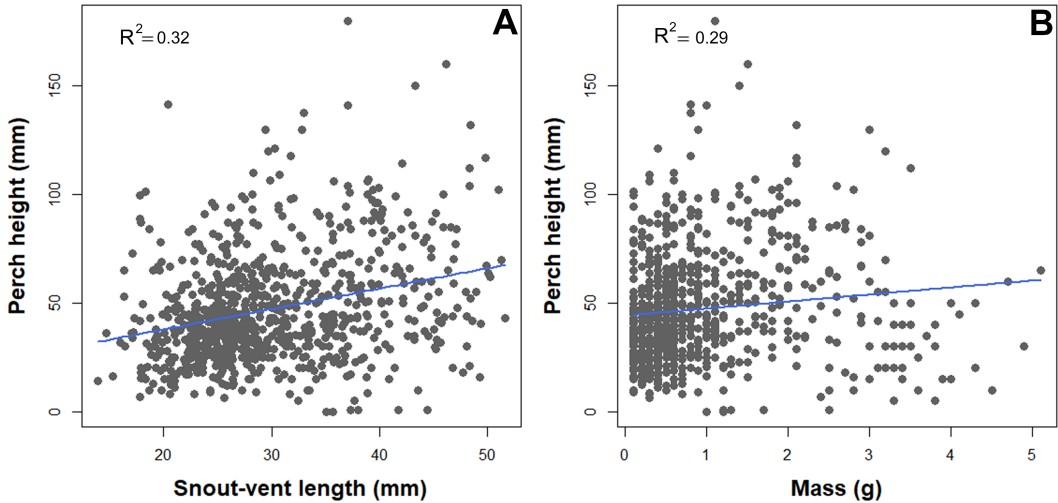

**Figure 2 Scatterplot depicting associations between perch high and natural history traits of *B. pandi*.** (A) A positive correlation between body length and perch high. (B) A positive correlation between weight and perch high.

distinguished by habitat variables such as vegetation layers, leaf litter depth and mean environmental temperature (Fig. 3B).

Results of the multiple regression analysis showed that leaf litter depth, mean environmental temperature, percent vegetation cover, and elevation were the variables that contributed the most to explaining the observed variability in the abundance of *B. pandi* ($r^2 = 60.3\%$, $P < 0.001$). These habitat variables also composed the best-fitted regression model (Table 4). However, this model showed a moderate autocorrelation (DW = 1.10), which means that the variance explained by the habitat variables may be close to 60.3%.

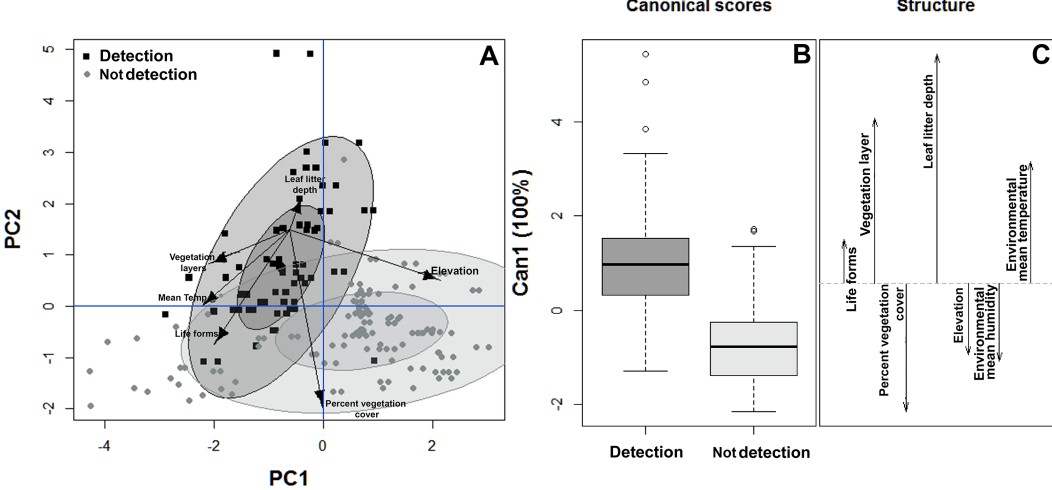

**Figure 3  Habitat suitability of *Bolitoglossa pandi*.** (A) Principal component analysis showing the observed variability in habitat structure attributes between plots with detection/non-detection of *Bolitoglossa pandi*. PC1 = First principal component (38.8%). PC2 = Second principal component (23.8%) . Black squares = detection of *Bolitoglossa pandi*. Grey dots = Non-detection of *Bolitoglossa pandi*. The inner ellipse represents 0.5 of significance; the outer ellipse represents 0.95 of significance. (B) Quadratic discriminant analysis depicting the habitat variables that have greatest discrimination capacity between quadrants where *Bolitoglossa pandi* was detected or non-detected. Length of the vector denotes discrimination capacity of each habitat variable.

**Table 3  Principal component correlation matrix.** The upper diagonal part contains correlation coefficient estimates. The lower diagonal part contains corresponding *p*-values. Bold values denote statistical significance at the *p* < 0.05 level.

| Habitat structure variables | Growth forms | Vegetation Layer | Percent vegetation cover | Leaf litter depth | Elevation (m) a.s.l | Environmental mean temp °C | KMO | BST |
|---|---|---|---|---|---|---|---|---|
| Growth forms | ***** | 0.481 | 0.263 | 0.002 | −0.384 | 0.368 | | |
| Vegetation layers | **<0.001** | ***** | −0.078 | 0.340 | −0.364 | 0.410 | | |
| Percent vegetation cover | **<0.001** | 0.285 | ***** | −0.274 | 0.045 | −0.061 | | |
| Leaf litter depth | 0.979 | **<0.001** | **<0.001** | ***** | 0.149 | 0.060 | 0.6 | **<0.001** |
| Elevation (m) a.s.l | **<0.001** | **<0.001** | 0.541 | **0.040** | ***** | −0.629 | | |
| Environmental mean temp °C | **<0.001** | **<0.001** | 0.401 | 0.409 | **<0.001** | ***** | | |

**Notes.**
KMO, Kaiser-Meyer-Olkin test; BST, Kaiser-Meyer-Olkin test.

## Activity pattern

*Bolitoglossa pandi* is completely nocturnal; its activity period extends throughout the night, from 18:30 h until 05:00 h. The environmental temperature recorded during this activity period ranged from 12.6–25.6 °C ($\bar{X}$ = 15.3), and relative humidity ranged from 73.8–98.8 ($\bar{X}$ = 80.3). The activity peaks were all subject to environmental influences showing a significant association with the local weather conditions. The observed salamander abundance was strongly and positively correlated with the environmental temperature ($R_{ET}$ = 0.251, $P < 0.001$), whereas it was moderately and negatively correlated with

**Table 4 Multiple regression models.** Akaike Information Criterion (AIC) employed to select the 'best model' that relates the fluctuation in habitat structure variables and the abundance of *Bolitoglossa pandi*. Dependent variable: Lntransformed abundance of B. pandi (Lnn). Independent variables: Per.veg.cov; Percent vegetation cover; LntransformedMean Temp, Environmental mean temperature; LnMeanH, Lntransformed relative humidity; Lnlife forms, Lntransformed number of vegetation life form; Veg.leyers, Vegetation layers; Lnleaf litter, leaf litter depth; Nor.test, KolmogorovSmirnov's test for normality; Hom.test, BreuschPagan test for homoscedasticity; and Aut.test, DurbinWatson test for autocorrelation. Values shown are standard error (SE) and *t* test-value. Bold values denote statistical significance at the *p* < 0.05 level.

| Multiple regression models | AIC | ΔAIC | Nor.test | Hom.test | DW |
|---|---|---|---|---|---|
| Lnn~Lnleaf.litter+LnMean.temp+Per.veg.cov+Elevation | −73.64 | 0.0 | | | |
| Lnn~Lnleaf.litter+LnMean.temp+Per.veg.cov+Elevation+LnGrowth.forms | −73.08 | −0.56 | | | |
| Lnn~Lnleaf.litter+LnMean.temp+Per.veg.cov+Elevation+LnMean.Hr | −71.92 | −1.72 | 0.45 | 0.06 | 1.10 |
| Lnn~Lnleaf.litter+LnMean.temp+Veg.leyers+Per.veg.cov+Elevation+LnMean.Hr+LnGrowth.forms | −70.04 | −3.60 | | | |

| The ''best'' multiple regression model | Relative importance for *B. pandi* abundance | Estimate | SE | *t*-value | P(>|*t*|) |
|---|---|---|---|---|---|
| Intercept | | 3.33 | 1.93 | 2.79 | **0.006** |
| Ln leaf litter | 64% | 0.88 | 0.071 | 12.34 | **<0.001** |
| Ln Mean temp | 18.5% | 0.69 | 0.167 | 4.12 | **<0.001** |
| Percent vegetation cover | 12% | −1.87 | 0.50 | −3.73 | **<0.001** |
| Elevation (m) a.s.l. | 5.5% | −0.001 | 0.0004 | −2.46 | **0.014** |
| *F* = 69.18, *df* = 4–182, *P* < 0.0001 | | | | | |

relative humidity ($R_{RH} = -0.174$; $P = 0.017$). Across sampling occasions, we observed two consistent activity peaks, the first from 20:30 h to 21:30 h, and the second from 23:30 h to 00:30 h (Fig. 4A). Results of the nested ANOVA indicated that salamander abundance between these peaks was significantly different ($F_{9-259} = 4.57$, $P < 0.001$; Fig. 5A, Table 5), being higher in the second activity peak during the second and third sampling occasions, with the opposite pattern during the first sampling occasion (Figs. 4B–4D).

Likewise, the observed body length of the salamanders showed significant differences between the two activity peaks ($K_{8df} = 99.70$, $P < 0.001$, Fig. 5B), suggesting niche time-partitioning between body size classes. Most of the salamanders observed during the first activity peak had SVL >30 mm (adults), whereas all the salamanders observed at the second activity peak had SVL <27 mm (juveniles and neonates). However, in contrast to the differences in observed abundance between activity peaks, the variation in salamander SVL was between peaks was consistent across the three sampling occasions.

## Population stage-structure

A total of 1391 individuals of *B. pandi* were observed throughout the study in the Supatá population, exhibiting a population density ranging from 0.04–1.44 individuals/m². The population stage-structure showed significant differences between sampling occasions, suggesting seasonal variability between development stages (Table 6). Regardless of the variability observed in the population stage-structure of *B. pandi*, the population is mostly dominated by juveniles and neonates which represent between 34–64% of individuals

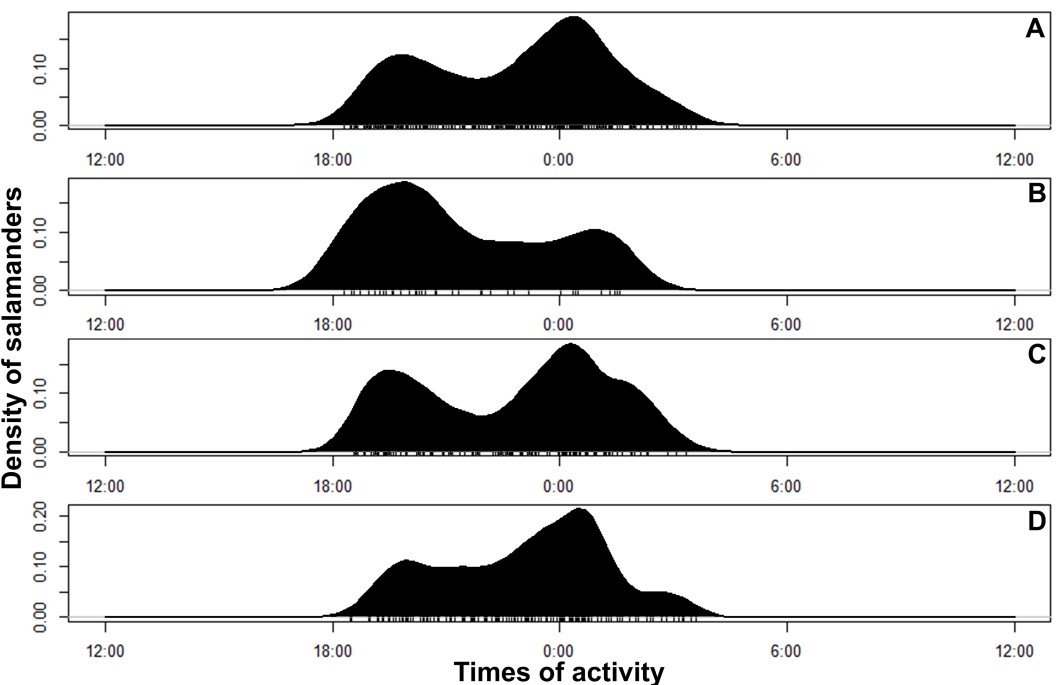

**Figure 4 Circular kernel density models showing overall activity patterns of *Bolitoglossa pandi*.** Bottom small vertical bars depicts the independent detections observed. (A) Activity pattern observed combining all sampling occasions. (B) Activity pattern observed in the first sampling occasion: low rainy season (from mid-September to mid-December 2017). (C) Activity pattern observed in the second sampling occasion: high rainy season (from mid-March through June 2017). (D) Activity pattern observed in the third sampling occasion: low dry season (from July to mid-September 2018).

(Fig. 6). Neonates were observed across all sampling occasions, indicating constant recruitment. During the second sampling occasion, a bias in the median SVL towards smaller individuals was observed, suggesting that the major recruitment peak occurs during March when the high rainy season start (Fig. 6B). However, no samples were available between October to mid-December when the low rainy season occurs, thus, a second recruitment peak could be possible observed during this period. On the contrary, adults were conspicuous throughout all sampling occasions, especially so in the first sampling occasion.

## Morphological variability and comparisons to other species

*Bolitoglossa pandi* is a small species, with SVL = $13.4-51$ mm ($N = 1034$), $34.4-48$ mm adult males ($n = 7$), and SVL $36.6 - 51$ mm adult females ($N = 8$). Extensively webbed hands and feet with third toes and triangular digits; ventral surfaces of digit tips without terminal flattened tubercles; snout short and rounded in the lateral profile; head length 4.9–10.1 mm; head width 5.1–7.9 ($N = 17$); snout rounded in dorsal view, irregular white spots, cream-colored nasolabial grooves, and edges of the lips irregularly dark brown with irregular light spotting (Figs. 7A–7B); protruding eyes on dorsal view, brown iris with black reticules (Fig. 7B); well-defined post-cephalic constriction; ventral surfaces (preserved)

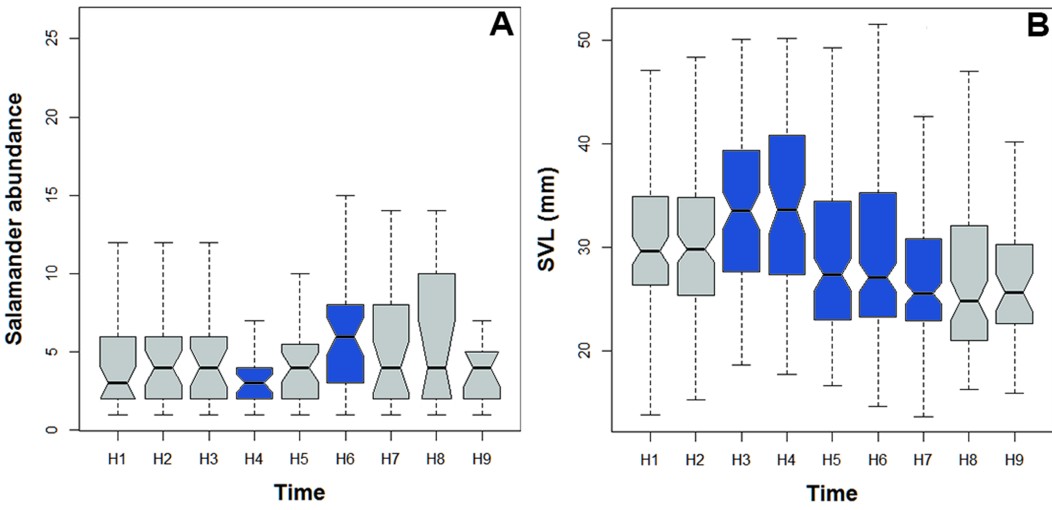

**Figure 5  Abundance and body size variability during the activity period of *Bolitoglossa pandi*.** Activity period of *Bolitoglossa pandi* was divided into nine time-intervals (from H1 = 18:30–19:30, H2 = 19:31–20:30, H3 = …until H9 = 02:31–03:30). (A) Nested ANOVA depicting the abundance variability among activity peaks observed. (B) Kruskal–Wallis test depicting niche partitioning between body size classes. Significant differences between time-intervals highlighting in blue.

**Table 5  Nested ANOVA results.** Standard error (SE). Bold values denote statistical significance at the $p < 0.05$ level.

| Time interval | Total salamanders observed across the study | Mean of salamanders observed per night sampled | Range of salamanders observed per night sampled | Estimate | SE | t value | P(>\|t\|) | Normality test | Levene test |
|---|---|---|---|---|---|---|---|---|---|
| H1 (18:30–19:30) | 154 | 4.16 | 1–16 | −0.0257 | 0.0234 | −1.085 | 0.279 | | |
| H2 (19:31–20:30) | 172 | 4.91 | 1–20 | 0.010 | 0.022 | 0.461 | 0.645 | | |
| H3 (20:31–21:30) | 123 | 4.73 | 1–12 | 0.013 | 0.023 | 0.525 | 0.599 | | |
| H4 (21:31–22:30) | 97 | 3.46 | 1–8 | −0.047 | 0.023 | −2.037 | **0.042** | | |
| H5 (22:31–23:30) | 158 | 4.39 | 1–16 | −0.006 | 0.020 | −0.309 | 0.757 | | |
| H6 (23:31–00:30) | 260 | 6.05 | 1–20 | 0.049 | 0.020 | 2.493 | **0.013** | | |
| H7 (00:31–01:30) | 196 | 4.73 | 1–18 | 0.040 | 0.022 | 1.834 | 0.067 | $P = 0.08$ | $P = 0.2$ |
| H8 (01:31–02:30) | 116 | 6.47 | 1–26 | 0.003 | 0.030 | 0.100 | 0.920 | | |
| H9 (02:31–03:30) | 62 | 4.43 | 1–26 | 0.003 | 0.030 | 0.100 | 0.920 | | |

brown or dark grey with numerous tiny cream guanophores (Figs. 7C–7D); inverted bracket shaped scapular spots (Fig. 7J); males have white testes.

**Table 6 Wilcoxon test results.** N = number of salamanders observed. Bold values denote statistical significance at the $p < 0.05$ level.

| Sampling occasion | Occasion 1 | Occasion 2 | Median | Neonates | Juveniles | Adults |
|---|---|---|---|---|---|---|
| Occasion 1 | – | – | 32.05 | $N = 4$ (8%) | $N = 13$ (26%) | $N = 33$ (66%) |
| Occasion 2 | $W = 11825$<br>**$P < 0.001$** | – | 25.40 | $N = 55$ (34%) | $N = 49$ (30%) | $N = 57$ (36%) |
| Occasion 3 | $W = 22590$<br>**$P = 0.006$** | $W = 84198$<br>**$P < 0.001$** | 28.20 | $N = 76$ (13%) | $N = 277$ (46%) | $N = 252$ (42%) |

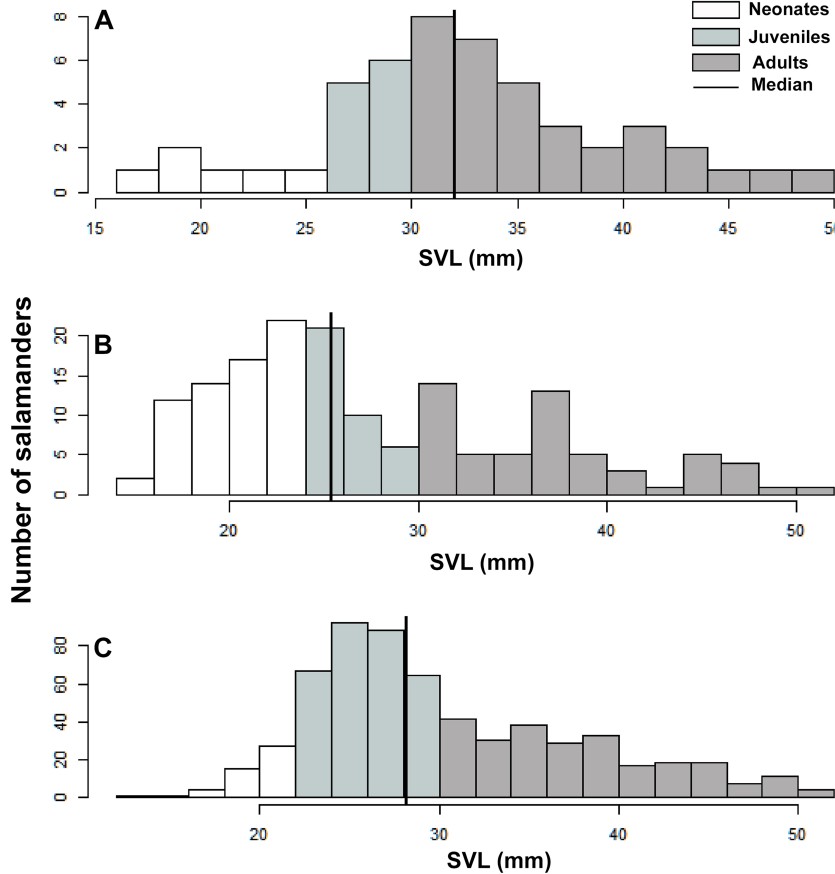

**Figure 6 Population stage-structure of the *Bolitoglossa pandi* at the Cuzcungos locality across the sampling period.** (A) First sampling occasion, September–October 2017. (B) Second sampling occasion, March 2018. (C) Third sampling occasion, July 2018.

*Bolitoglossa pandi* can be distinguished from its Colombian congeners by the extensive webbing of its hands and feet (versus webbing of hands and feet reduced in *B. adspersa, B. hiemalis, B. hypacra, B. palmata, B. ramosi, B. savagei, B. tamaense, B. tatamae, B. walkeri* and *B. vallecula*). Furthermore, it can be distinguished from species with extensive webbing (such as *Bolitoglossa lozanoi* and *B. nicefori*) by having more protruding eyes, and a longer and triangular third digit. *Bolitoglossa pandi* also differs from *B. biseriata* and *B. silverstonei*

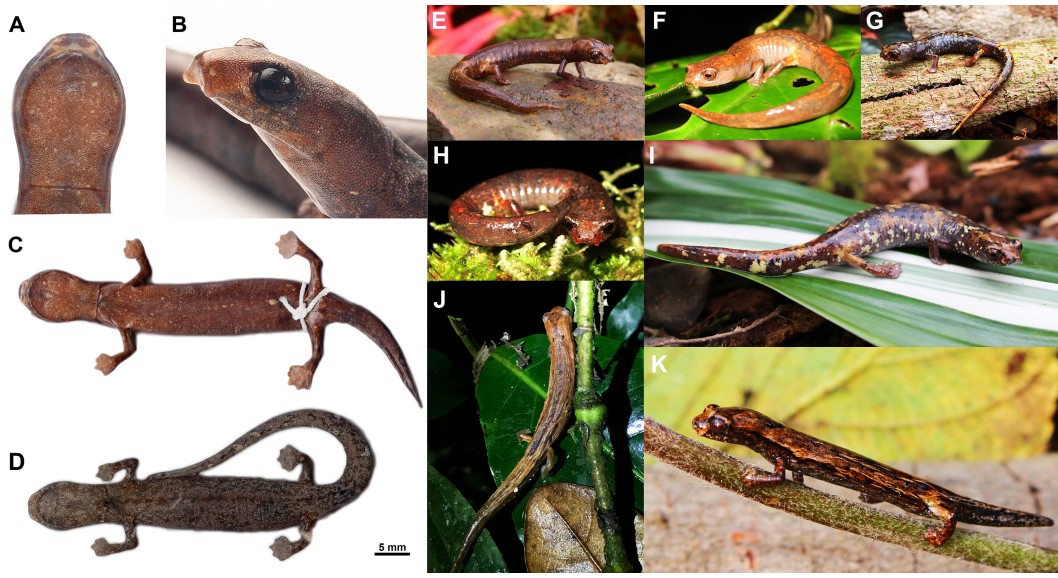

**Figure 7 Color variability observed in living and preserved specimen of *Bolitoglossa pandi*.** (A) ventral surface of the head (ICN 45000; Pandi, Cundinamarca). (B) Lateral view of the edges lip showing the color pattern irregularly dark brown with irregular light spotting (ICN 58501, in life. Supatá, Cundinamarca). (C) Ventral surfaces uniformly dark brown or dark grey with some irregular white circular spots (ICN 45000; Pandi, Cundinamarca. ICN 58502, Supatá, Cundinamarca). (D) Uniform dorsal color pattern. (E) Dorsal color reddish-brown that can have diffuse grey or dark blotches scarcely distinguishable to the dorsolateral. (F–H) Dorsal surfaces exhibiting an ochre pattern, with some diffuse irregular dark brown, yellow or cream spots. (I–J) Dorsolateral surfaces diffuse band in the paravertebral region covering almost the entire dorsal surface or includes a dark brown inverted triangle shape in the interorbital region. Pictures by: Teddy Angarita-Sierra.

by having a dark brown or dark grey ventral surface with irregular white dots (versus cream ventral surface with brown suffusions and dots in *B. biseriata* and *B. silverstonei*), brown-reddish iris with black reticles (versus golden with brown dots in *B. biseriata* and *B. silverstonei*), and white testes in adult males (versus black mesorchium in testes of adult males of *B. biseriata*). It also differs from *B. medemi* by the absence of digital depressions in the tips on the digits and toes (present in *B. medemi*). *B. pandi* also presents an upper lip with irregular light spotting (versus uniform in *B. guaneae*). *B. pandi* can be differentiated from *B. altamazonica* and *B. leandrae* by having extensive interdigital webbing with a longer and triangular third digit (complete webbing and tips rounded in *B. altamazonica* and *B. leandrae*). *B. pandi* is morphologically very similar to *B. phalarosoma,* but it differs by having a dark brown or dark grey ventral surface with diffuse white dots and some white blotches (versus usually light brown ventral surfaces, with some irregular cream spots in *B. phalarosoma*), plantar and palmar regions in ventral dark brown or dark grey view (versus cream in *B. phalarosoma*). *B. pandi* also differs from *B. capitana* by having a smaller adult size with a longer and triangular third digit (versus rounded third digit in *B. capitana*) and a shorter head.

## Color variability

*Bolitoglossa pandi* exhibits a wide variation in color, ranging from a uniform dorsal color pattern of different shades of light to dark brown (Fig. 7E); one reddish-brown pattern that can have diffuse grey or scarcely distinguishable dark blotches that extend to the dorsolateral region (Fig. 7F), or an ochre pattern, with some diffuse irregular dark brown, yellow or cream spots (Figs. 7G–7I). Some specimens exhibit a very diffuse band in the paravertebral region covering almost the entire dorsal surface or including a dark brown inverted triangle shape in the interorbital region (Figs. 7J–7K). The caudal region is highly variable, ranging from uniform reddish-brown or completely ochre segments, with very small scattered white dots (Figs. 7D–7F) to irregular cream, yellow or orange patches (Fig. 7I), and very small or irregular longitudinal black spots (Figs. 7I–7K). The distal end of the tail becomes uniform light brown and has cream blotches towards the proximal region in some individuals (Fig. 7H). The cephalic region is dark brown with some white spots with irregular dots up to the supralabial region in lateral view (Fig. 7B); a loreal region with ochre patches and light brown iris with black reticles; nasolabial projections are cream; the ventrolateral surface is dark brown. Ventral surfaces are dark brown with some cream and white blotches with scattered dots; the mental and gular surfaces are uniformly dark brown or dark grey with some irregular white circular spots bordering the maxillary region. In adult males the mental gland is light brown; the palmar and plantar surfaces are always dark brown or dark grey.

## DISCUSSION

The new localities add to the known range of *Bolitoglossa pandi* and the detailed examination of the Supatá population allowed not only the expansion of knowledge about the distribution and natural history traits of this species but also the reassessment of its conservation status and the validation of its characters for taxonomic identification vis-a-vis its congeners. Despite efforts done by herpetologists who sought to characterize and describe the Andean amphibians in the second half of the 20th century and the beginning of the 21st (*Ruiz-Carranza, Ardila-Robayo & Lynch, 1996*; *Lynch, Ruiz-Carranza & Ardila-Robayo, 1997*; *Lynch, 1999*; *Arroyo, Jerez & Ramírez-Pinilla, 2003*; *Acosta-Galvis, 2015*; *Acosta-Galvis, 2020*), large areas of the western slope of the Cordillera Oriental of Colombia still lack intensive sampling. Therefore, the known distribution of *B. pandi* as well as other Andean amphibians is still fragmentary.

Since *Brame & Wake*'s (*1963*) original description of *B. pandi,* several characters such as dorsal surface color (Fig. 7) have had conflicting or ambiguous diagnostic characters (*Acosta-Galvis & Gutiérrez-Lamus, 2012*). However, our findings increased the understanding of the morphological variability of *B. pandi.* Based on living specimens obtained in the municipalities of Guaduas, Supatá, and Villeta, we add new data to the original description (*Brame & Wake, 1963*) because these had not been described in life (and the holotype is poorly preserved), and evidence of a broad intrapopulation color variation was not available due to restricted sampling. Our findings allow clarification of taxonomical misidentifications in the literature. For example, *Acosta-Galvis & Rueda-Almonacid (2004)*,

during the first threat assessment of *B pandi,* included erroneously a picture of *B. walkeri* as the species' portrait.

Neotropical salamanders have been considered amphibians with secretive habits and low encounter rates into the Andean Forest (*Brame & Wake, 1963*; *Gibbons, 2013*; *Barrio-Amorós & Fuentes-Ramos, 1999*; *Acosta-Galvis & Gutiérrez-Lamus, 2012*). Nevertheless, our results challenge this general assumption and provide support for previous studies which related specific environmental conditions with high abundance and density of Andean salamander populations (*Houck, 1977*; *Jimenez, 1994*; *Salgado-Aráuz, 2005*; *Cadenas et al., 2009*; *Ortega, Monares-Riaño & Ramírez-Pinilla, 2009*; *Gutiérrez-Lamus, Lynch & Martínez-Villate, 2011*; *Neckel-Oliveira et al., 2011*; *Del Río-García, Serrano-Cardozo & Ramírez-Pinilla, 2014*; *Cruz, Galindo & Bernal, 2016*). *Bolitoglossa pandi* follows the common pattern observed in upland tropical salamanders, having a narrow elevational range (1,650–2,315 m. a.s.l), in which elevation, leaf litter depth (>6 cm), vegetation layers (>5 vegetation strata), and environmental mean temperature (16–19 °C) were the main predictor variables for their presence, abundance and population density (*Wake & Lynch, 1976*; *Gutiérrez-Lamus, Lynch & Martínez-Villate, 2011*; *Cruz, Galindo & Bernal, 2016*; *Donaire et al., 2019*). Likewise, observed patterns of habitat use by *B. pandi* agree with those previously reported by *Del Río-García, Serrano-Cardozo & Ramírez-Pinilla (2014)* in which the salamanders exhibiting both arboreal and terrestrial habits, perching on the upper side of leaves (at heights between 2.5 and 250 cm above the forest floor), showing a positive correlation between perch height, SVL and mass. Additionally, the highest abundance of neonates and juveniles observed during high rainy season (from mid-March through June) agrees previously observation made by these authors. However, a negative correlation or no correlation between perch height and salamander body size has been reported for *B. paraensis* and *B. nicefori*, respectively, suggesting that climatic variability or elevational range distribution could determine microhabitat use (*Ortega, Monares-Riaño & Ramírez-Pinilla, 2009*; *Simões Correa & Chagas-Rodriguez, 2017*). This and many more questions concerning microhabitat use of South American *Bolitoglossa* salamanders remain open, highlighting the fact that the state of knowledge on their ecology and natural history still presents many gaps.

The occurrence of individuals of *Bolitoglossa pandi* was strongly related with habitat structure. Sampling plots in which *B. pandi* was present were positively correlated with highly structured habitats and deep leaf litter, characteristics traditionally associated with suitable habitats for amphibians because they provide food (leaf litter arthropods), shelter, nesting sites, and microclimate stability by retaining moisture through the soil interface after rainfall events (*Heatwole, 1962*; *Jaeger, 1980*; *Harvey-Pough et al., 1987*; *Vitt & Caldwell, 2001*). Additionally, the leaf litter depth was significant correlated with percent vegetation cover and vegetation layers, which agrees with the pattern previously reported by *De Maynadier & Houlahan (2008)* in which the composition and structure of the leaf litter and local tree canopy is significantly related in tropical forests.

Conversely, highly homogenous sampling plots in which were dominant pastures, graminoids, bare soil, and rocks, showed the lowest or no detection grade of *B. pandi*. Therefore, this result showed that habitat degradation due local human activities such

logging, and cattle ranching have direct negative effects in the quality of leaf litter and general habitat structure of *B. pandi* (*Vitt & Caldwell, 2001*). The detection and high local abundance of *B. pandi* was strongly correlated with the same habitat variables associated with the detection of other Andean salamanders such as *B. adspersa, B. altamazonica, B. nicefori,* and *B. orestes,* and *B. ramosi* (*Valdivieso & Tamsitt, 1965*; *Wake & Lynch, 1976*; *Cadenas et al., 2009*; *Gutiérrez-Lamus, Lynch & Martínez-Villate, 2011*; *Leenders & Watkins-Colwell, 2013*; *Galindo, Cruz & Bernal, 2018*). Thus, the strong dependency on a narrow environmental range of conditions makes the Andean bolitoglossines highly vulnerable to local changes by human activities affecting these habitat variables.

Usually, activity in bolitoglossines has been associated with variation in relative humidity or climatic conditions, as well as the breeding season (*Vial, 1968*; *Wake & Lynch, 1976*; *Ortega, Monares-Riaño & Ramírez-Pinilla, 2009*; *Simões Correa & Chagas-Rodriguez, 2017*. Activity of *Bolitoglossa pandi* follows this general pattern but with certain deviations compared to some of its congeners. For example, the number of *B. pandi* active individuals was strongly and positively correlated with the environmental temperature, whereas it was moderately and negatively correlated with the relative humidity (versus a positive correlation between active individuals and relative humidity observed in *B. mombachoensis, B. paraensis,* and *B. subpalmata* (*Vial, 1968*; *Salgado-Aráuz, 2005*; *Simões Correa & Chagas-Rodriguez, 2017*)). This correlation was consistent across all sampling occasions, suggesting that climate variability has little influence on the activity pattern of *B. pandi*. However, future studies should be carried out to clarify which environment variables can determine the general activity pattern in the tropical *Bolitoglossa* species.

Niche partitioning has been reported for many populations of Nearctic and Neotropical salamander species (*Jaeger & Gergits, 1979*; *Wicknick, 1995*; *Arif, Adams & Wicknick, 2007*; *Jaeger et al., 2016*). Intraspecific niche partitioning in salamanders has been explained as a life history strategy to maximize foraging success, predator avoidance, and mating success (*Jaeger & Gergits, 1979*; *Holomuzki, 1986*; *Cloyed & Eason, 2017*). Particularly, intraspecific niche partitioning due to ontogenetic shifts or sexual microhabitat selection has been documented for some *Bolitoglossa* species (e.g., *B. dofleini* and *B. nicefori* (*Raffaëlli, 2007*; *Ortega, Monares-Riaño & Ramírez-Pinilla, 2009*)). Previously, *Del Río-García, Serrano-Cardozo & Ramírez-Pinilla (2014)* explored whether intraspecific niche partitioning due to ontogenetic shifts or sexual microhabitat selection was present in *B. pandi* populations at the municipality of Supatá. However, these authors did not find significant differences in microhabitat use between sexes or body size classes. However, our study showed that adults and juveniles+neonates of *B. pandi* from the Cuzcungos locality showed significant differences between the activity peaks, indicating that development stages of this species are disaggregated through the niche time-axis. Future experimental studies will be needed to understand how intraspecific niche time-partitioning is acting through environmental heterogeneity in the ecology and evolution of the South American bolitoglossines.

## CONCLUSIONS

*Bolitoglossa pandi* followed the expected association between habitat structure and morphological traits predicted by the general adaptive radiation pattern observed in Neotropical salamanders (*Darda & Wake, 2015*). Habitats with a high percent of vegetation cover and leaf litter depth provide suitable environmental conditions for this species and define the sets of predictor variables associated with occupation of *B. pandi* in the Andean forests. Given the narrow elevational range of this species, local changes of these habitat variables along small areas could lead to extirpation of their populations. We recommend conducting occupancy modelling as a tool for monitoring the detection/non-detection of *B. pandi* related with habitat loss.

Urodela is the most endangered vertebrate group on Earth being more than 54.7% of its species are thought of as "endangered" (*Hernandez, 2016* ). Our result agrees with the general pattern observed in other salamander taxa in which habitat disturbance driven by human activities has deleterious effects on their presence or population densities given their high habitat specificity (*Lips, 1998*; *Collins & Storfer, 2003*; *Hernandez, 2016*). Habitat loss continues to stand out as the main threat for *B. pandi*. This fact has significant importance in conservation issues because the Andean forest is one of the most threatened ecosystems by human activities in Colombia (*Etter et al., 2018*). Therefore, despite our observations of a healthy and stable salamander population that showed high population density and constant recruitment in the municipalities of Supatá, Guaduas, and Villeta, we recommend that *B. pandi* be retained as Endangered (EN) on the IUCN Red List based on the IUCN Criterion B, given its restricted extent of occurrence (ca. 2,500 km$^2$), as well as the ongoing habitat loss within its range due to agriculture, cattle ranching, logging, and urban development.

Finally, Colombian herpetologists must encourage private and public research centers, universities, conservation agencies and industries to support fieldwork that seeks to increase the knowledge on amphibian diversity, filling the distribution gaps of the Andean species, particularly salamanders (*Ospina-Sarria & Angarita-Sierra, 2020*). The latter are essential to develop successful conservation and management strategies and environmental regulations in a megadiverse country such as Colombia (*Bury, 2006*). An example is this study, wherein, we characterized several important aspects of population biology and ecology of the endangered species *B. pandi*. By filling gaps in our knowledge of this species, we were able to describe in detail the habitat requirements for its conservation and provide an update of its conservation status.

## ACKNOWLEDGEMENTS

We thank all students of the Grupo de Investigación de Biología de Organismos Tropicales (BIOTUN) of the Universidad Nacional de Colombia for their help and support during the fieldwork and lab procedures. We especially thank Maria Carolina Becerra (Biology department, UNAL) for her friendship and support through all this study; the staff of the NGO YOLUKA for their general support; Rebeca Morantes-Zamora for her help with the map; John D. Lynch (Instituto de Ciencias Naturales de Universidad Nacional de

Colombia) for his support, advice, and for allowing us to use the lab equipment under his care. Special thanks to Adriana Zuleta and Alejandro López for giving us the support and freedom to seek salamanders at Cuzcungos Natural Reserve; William Gutierrez Moreno, coordinator of the Project Management Program of Universidad Nacional de Colombia for his support. We thank Kelsey Neam (IUCN SSC Amphibian Specialist Group, Global Wildlife Conservation) and Thibaud Aronson (Missouri Botanical Garden) for the review of the manuscript. We thank the staff of Ecotropico, as well as to Rafael Enrique Martínez and family, José Adelfo Vargas and family for kindly allowing us to seek amphibians and reptiles at their forest and farmlands. Special thanks to all the community of the municipalities of Guaduas, Supatá, Venecia, and Villeta to allow us to work together for the amphibian and reptile conservation of Colombia. Finally, thanks to Axel Hernandez and the anonymous reviewer for their valuable comments on this paper.

### Funding

This work was supported by Mohamed bin Zayed Species Conservation Fund (No. 172515606), The Rufford Foundation (No. 24220-1), Universidad Nacional de Colombia Agreement (Project 38615) and YOLUKA ONG, Fundación de Investigación en Biodiversidad y Conservación (Act. 2017-01). The funders had no role in study design, data collection and analysis, decision to publish, or preparation of the manuscript.

### Grant Disclosures

The following grant information was disclosed by the authors:
Mohamed bin Zayed Species Conservation Fund: 172515606.
The Rufford Foundation: No. 24220-1.
Universidad Nacional de Colombia Agreement: 38615.
YOLUKA ONG.
Fundación de Investigación en Biodiversidad y Conservación: 2017-01.

### Competing Interests

The authors declare there are no competing interests.

### Author Contributions

- Teddy Angarita-Sierra conceived and designed the experiments, performed the experiments, analyzed the data, prepared figures and/or tables, authored or reviewed drafts of the paper, achieving that the community give the permits to develop fieldwork, and approved the final draft.
- M. Argenis Bonilla-Gómez performed the experiments, authored or reviewed drafts of the paper, providing the lab facility, and approved the final draft.
- David A. Sánchez conceived and designed the experiments, performed the experiments, authored or reviewed drafts of the paper, field work coordinator, achieving that the community give the permits to develop fieldwork, and approved the final draft.

- Andres R. Acosta-Galvis performed the experiments, analyzed the data, prepared figures and/or tables, authored or reviewed drafts of the paper, and approved the final draft.
- Hefzi Medina-Ovalle, Anggi Solano-Moreno, Simon Ulloa-Rengifo, Daniela Guevara-Guevara, Juan J. Torres-Ramirez and Sebastián Curaca-Fierro performed the experiments, prepared figures and/or tables, database compilation, and approved the final draft.
- Diego M. Cabrera-Amaya performed the experiments, prepared figures and/or tables, authored or reviewed drafts of the paper, and approved the final draft.
- Jhon A. Infante-Betancour performed the experiments, prepared figures and/or tables, authored or reviewed drafts of the paper, providing the field work equipment, and approved the final draft.
- Luisa F. Londoño-Montaño performed the experiments, prepared figures and/or tables, providing the field work equipment, and approved the final draft.
- Diana X. Albarán-Montoya and Lesly R. Peña-Baez performed the experiments, prepared figures and/or tables, and approved the final draft.

## Animal Ethics

The following information was supplied relating to ethical approvals (i.e., approving body and any reference numbers):

This study was conducted in accordance with Colombian Animal Welfare Law and Collection of Wild Specimen's Biological Diversity Acts (Ley 1774, 2016; Decreto 1376, 2013), as well as considerations of the Universal Declaration on Animal Welfare (UDAW) endorsed by Colombia in 2007. Research on live salamander was allowed under ethical approvals reference number 38615 issued by the National University of Colombia.

## Field Study Permissions

The following information was supplied relating to field study approvals (i.e., approving body and any reference numbers):

Fieldwork was done under a Collection of Wild Specimen's Biological Diversity Non-commercial Purpose of Scientific Research Permit issued by the National University of Colombia (Research Project 38615), and the Colombian National Environmental Licensing Authority (ANLA) by resolution No. 0255 of 14 March 2014. All specimens of *Bolitoglossa pandi* collected were deposited in the amphibian collection at Instituto de Investigación de Recursos Biológicos Alexander von Humboldt, (IAvH-Am 10303-10308), as well as in the amphibian collection at the Instituto de Ciencias Naturales de la Universidad Nacional de Colombia (ICN 58493-58503).

## Data Availability

The raw measurements are available in the Supplementary Files.

All *Bolitoglossa pandi* specimens were deposited in the amphibian collection at Instituto de Investigación de Recursos Biológicos Alexander von Humboldt, (IAvH-Am 10303-10308), as well as in the amphibian collection at the Instituto de Ciencias Naturales de la Universidad Nacional de Colombia (ICN 58493-58503).

## Supplemental Information

Supplemental information for this article can be found online at http://dx.doi.org/10.7717/peerj.9901#supplemental-information.

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
