# Peer review of "Distribution, habitat suitability, conservation state and natural history of endangered salamander Bolitoglossa pandi"

_PeerJ, doi:10.7717/peerj.9901_

## Round 0.1 · original submission · Minor Revisions

Thank you for submitting your manuscript to PeerJ. I have sent your paper to expert referees for their consideration. I have now received their comments back and have read through your paper carefully myself. Enclosed please find the reviews of your manuscript.
The reviews are in general favourable and suggest that, subject to minor revisions, your paper could be suitable for publication. Please consider these suggestions, especially the very detailed review of the first refferee, and I look forward to receiving your revision.
Please note that this reviewer notes that the paper has to be proofread by an English native-equivalent-speaking colleague or by manuscript proofread service.
I personally would appreciate if you would consider adding a thumbnail showing your focus species to Fig. 1 (the map) and possibly some other graphs to make them look more attractive for a potential reader.

Reviewer 1 ·

Basic reporting

In general, this manuscript is well written but I have noted numerous grammatical and wording changes that need to be made. While I do not know if any of the authors are native English speakers, and while the quality of the English is very high throughout the manuscript, it would benefit from having a native speaker revise it briefly (if none of the authors are native speakers) to pick up on odd phrasings that I have not noted. I tried to note them all but became less diligent about it toward the end of the manuscript. Finally, the literature cited has numerous formatting errors, which I did not note specifically because it appears that this section has not yet been formatted carefully for the journal.

In Figure 1, what is the orange color in the inset map? Does it correspond to the department of Cundinamarca? If so, this should be stated. It would be good to put a box around the area in the inset that corresponds with the bigger map. In Figure 2, it is almost impossible to determine what the relation of perch height is with the other two variables because there are simply too many points on the plot. Because mass and SVL are correlated for obvious reasons, I recommend plotting perch height against only one of these variables if there is a relationship between perch height and either mass or SVL. Figure 4 is also of somewhat low quality and doesn’t add much to the description in the text; I recommend deleting this figure. I don’t think that both Figures 8 and 9 are necessary, as they both show variation in color pattern. I recommend either combining them or deleting one.

Experimental design

This paper has a well-structured sampling design and seeks to provide data on several important aspects of population biology and ecology of a little-known species of salamander. It clearly fills a gap in our knowledge of this species and contributes to our understanding of South American Bolitoglossa, for which relatively little ecological or population information is available.

Validity of the findings

There are some statements of novelty and impact (which I tried to note in my specific comments) that are inappropriate for the format of this journal. Some aspects of the discussion and conclusions are not firmly based on the findings (see below). My main concern with the paper is in the discussion, which often strays from conclusions that can be drawn directly from the data. We lack these kinds of data for almost all species of Neotropical salamanders, and I think that the discussion and conclusions would be strengthened by really focusing on what can be concluded from the data available about the species and its natural history. The results section is a little odd as well, because it mixes primary data with what is effectively a re-description of the species. I suggest focusing more on the primary data, but this could be worked into a formal re-description if the authors desire.

Additional comments

This paper presents a lot of very valuable new information on abundance over time and space, as well as morphological and color variability, for a salamander that has been poorly known and little studied since its discovery. I give specific comments below by line number.

Line 36: Change n-dash to comma between Cundimamarca and Colombia

Line 50: Add “population” after “highly dense”

Line 69: Change “Bolitoglossa genus” to “genus Bolitoglossa”

Line 72: Change “Plethodontidae” to “plethodontid”. Additionally, Bolitoglossa is the most diverse and widespread genus of salamander in general, if the authors want to expand this statement beyond the Neotropics.

Line 73: AmphibiaWeb lists 133 species of Bolitoglossa, so this should be updated. It’s better to cite an online database like AmphibiaWeb because any publication on diversity numbers in the Neotropics is almost immediately out of date given the high rate of species descriptions.

Line 77: I don’t think it’s grammatically correct to say “high morphological crypsis”. You could change this to “high level of morphological crypsis” or “highly conserved morphology”.

Line 82: delete “so”

Lines 82–88: I don’t think that this statement is accurate. Almost all modern descriptions of species (including those dating back decades) include information on microhabitat use and habitat where species occurs. Books not cited here like the Amphibians and Reptiles of Costa Rica (Savage) or the Amphibians of Honduras (McCranie and Wilson) include detailed information on habitat and microhabitat use. There are also papers (Wake, 1987) with a lot of information on microhabitat use by Bolitoglossa.

Line 102: Change “nearby” to “near”. Also, if the exact type locality is unknown, how can this specimen be said to be from near the type locality?

Line 147: Change “at the municipalities” to “in the municipalities”

Line 155: I don’t know what is meant by “grove pastures”

Line 160: Insert space between “measured” and “its”

Lines 164–165: Were the live salamanders examined to see if they had a mental gland? This is typically a fairly reliable way to identify sexually mature adult males, although I am unfamiliar with the mental gland of this species.

Line 182: Change “estimatioof” to “estimation
Lines 198, 210: Change “vegetation percentage cover” to “percentage vegetation cover”

Line 201–202: I understand the logic here, but this analysis will not tell you what factor is responsible for the difference between detection and non-detection. The actual factor responsible could be correlated with one you measured but not actually in the dataset.

Line 206: Change “non-detected” to “not detected”

Line 210: This is a minor point, but here and elsewhere I would change “altitude” to “vegetation”. Altitude typically refers to the distance above ground (like the altitude at which planes fly).

Line 231: What exactly is meant by “population structure”? This phrase has different meanings in different disciplines (like ecology and population genetics) and it would be helpful to define it (i.e. as population age structure).

Line 236: change “nine-time intervals” to “nine time intervals” or “nine time-intervals”

Line 238: Change “employed” to “used”

Lines 247–251: Did these publications define these age categories for this species, or in general? I am not very familiar with South American Bolitoglossa, but there are very few species with sexually mature adults at 30 mm SVL. It would also be important to distinguish between males and females because males tend to mature at a smaller size than females.

Line 261: Change “in” to “at”

Line 268: I don’t think it’s grammatically correct to begin a sentence with “Also”. You could say “Additionally” instead.

Lines 272–273: I don't think that “natural history” is the right way to describe these traits. I suggest “morphological traits” or “morphology” instead.

Line 280: Instead of “explain”, why not say “are significantly associated with”? Explain implies that they completely explain the detection of salamanders, unless it is qualified (i.e. “partially explain”).

Line 282: I would only report values to one decimal place to not imply greater precision than there actually is in the analysis.

Line 284: Change “spaces” to “space”

Line 285: I recommend changing “strongly differentiated” to “differentiated”, given that they do overlap.

Line 286: Change “quadrats highly structure” to “highly structured quadrats”

Line 287: Remove “a” before “depth” or change to “with deep leaf litter”


Line 293: Change “environmental mean temperature” to “mean environmental temperature”

Line 296: The methods section states that an autocorrelation between 1 and 3 is moderate and this value is very close to one. If it were only 0.1 less, it would be “strong autocorrelation”. I think this sentence needs to be modified to reflect that there is moderate autocorrelation.

Line 301: Change comma after “nocturnal” to semicolon.

Line 302: Either say “beginning at… and ending at” or from…until…”.

Line 307: If I am interpreting this correctly, it should be rephrased as something like “whereas it was moderately and negatively correlated with relative humidity”

Line 308: “Despite” isn’t the right word here. Maybe “Across sampling occasions”?

Line 328: What is meant by “replacement rate” here?

Line 332: Delete “notorious” to remove value judgement

Line 339–340: There is literally no way that there are 13.4 mm adult males. This would be smaller than the smallest adults of Thorius, which are the very smallest salamanders. Furthermore, the size variation in Figure 7 implies that all adults measured were >30 mm. The proper way to determine if the specimens are adults or not is to assess sexual maturity either via examination of the mental gland, chloacal morphology, or internal reproductive organs. It’s fine to divide individuals into rough size classes to look at population age structure, but these data can’t then be used to make statements about adult size.

Line 339: This section is titled “morphological variability’ but is in fact basically a redescription of B. pandi. I suggest that the authors either reformat this section as a redescription based on additional material or limit themselves to describing the morphological variability without providing a diagnosis for the species.

Line 358: All of the species of Neotropical salamanders of which I am aware undergo an ontological change from white to black testes. The fact that the authors state that B. pandi has white testes makes me suspect further that many of the individuals they examined were not adults, although it is possible that this species retains white testes into sexual maturity. The appropriate way to determine this would be to examine the mental gland of the male specimens collected to determine if they are, in fact, sexually mature adults.

Lines 370–371: Change “less head width” to “shorter head”

Line 374: What does “wide polychromatic variability” mean? I think it would be better to just say “wide variation in color”.

Line 376: Change “diffused” to “diffuse” I can’t tell whether this should be rephrase as “dark blotches scarcely distinguishable from the dorsolateral region” or “scarcely distinguishable dark blotches that extend to the dorsolateral region”.


Line 379: “Exceptionally” implies a value judgment. Sah “very” instead.

Line 388: Change “creams” to “cream”

Line 396: Change “added to Bolitoglossa pandi distribution” to “added to the distribution of Bolitoglossa pandi”

Line 400–404: Bolitoglossa pandi is still a narrow endemic, occurring within only one department of Colombia. I don’t quite understand what is meant by this. Furthermore, what does “lack of comparisons [sic] with the relevant type material” mean in this context? Were salamander specimens available from this area but not assigned to B. pandi? I also don’t understand what is meant be the last part of the sentence about incomplete knowledge of morphological variability of Bolitoglossa. We know an enormous amount about the morphological variability of Bolitoglossa, and this manuscript really don’t have much to do with that. Rather, it improves our knowledge of one species of Bolitoglossa through the addition of detection data and morphology. I think that the real contribution of this paper could be stated much more accurately and concisely without making it something it isn’t.

Line 408: Delete “significantly”, as it implies a value judgment.

Line 409: The color data presented here don’t contrast with the original description because it presented no data on color in life. Rather, the new data add to the original description.

Line 412: “Polychromatism” seems like an unnecessarily complex (and partially inaccurate” word to describe color pattern variation.

Lines 418–423: I don’t quite understand what is meant by this sentence.
“Inconspicuous” just means that they are hard to see or notice. Is the idea that they are more abundant than previously suspected? If so, does this account for species like Bolitoglossa adspersa that are (or at least were) quite abundant at many localities?

Line 426: Change “arise” to “were” and remove quotes around “best”

Lines 427–431: I don't understand what is meant by this section either. Two of the references cited are for different, individual species of Bolitoglossa, whereas this sentence makes it seem like the results were expected based on what we already know about this species or about the genus in general. Bolitoglossa is extremely variable; some species are entirely terrestrial, in contrast to the “arboreal habits” listed here. I recommend deleting this entire sentence or at the very least rephrasing it so that it says something accurate that adds to the discussion.

Line 445: Change “promoting” to “were associated with”

Line 449: This is patently untrue. There is published information on females of Bolitoglossa dofleini being terrestrial while males are arboreal, and this is true of other species as well. Furthermore, it’s not clear to me which of the results presented here suggest that there is intraspecific habitat partitioning.

Conclusion: The conclusion section seems based on only one aspect of the results (the conservation status of the species). There is a lot of really important information on life history and abundance presented here and I suggest refocusing this section on what is directly able to be concluded from the data presented.

Figure 5: This figure is odd, because it seems like the same peak is repeated on both sides of each panel. It would be better to cut off the figure at 0:00 and 24:00 for clarity of interpretation.

·

Basic reporting

The paper is clear, only some mistakes regarding the english used. The English language should be improved to ensure that an international audience can clearly understand your text.
Please correct it.
In the Background section, please add the description for the microhabitat use in Bolitoglossa pandi.

Experimental design

Methods are sufficient, but please add Heyer et al. for the visual encounter surveys (VES):
See Heyer, R., Donnelly, M. A., Foster, M., & Mcdiarmid, R. (Eds.). (2014). Measuring and monitoring biological diversity: standard methods for amphibians. Smithsonian Institution.
And explain clearly Why did you choose this method ?
(not disturb the target sp. etc...)

Validity of the findings

All underlying data have been provided; they are robust, statistically sound, & controlled.
Please add in the discussion part, a comparison with other Bolitoglossa's habitat use;
and describe in details their habitat needs before conservation section.

Additional comments

This paper describes some new natural history observations on one species of threatened Bolitoglossine in Colombia, Bolitoglossa pandi. Although the methods are fairly routine and the sample sizes are small, this species is a part of a group of considerable conservation interest and the data presented do therefore provide an important addition to our understanding of their distribution, natural history and conservation.
However, many interesting findings in your data to underline clearly especially through the discussion part:
Describe in details Habitat + microhabitat use by B. pandi in comparison to other Bolitoglossines.
If you show its ecological specificities it's better in terms of conservation perspectives. This last point is really important to ensure your paper as a reference in ecology for Colombian salamanders.

---

## Round 0.2 · Minor Revisions

Thank you for submitting your manuscript to PeerJ. I have sent your paper to two expert referees for their consideration. I have now received their comments back and I can see that they are satisfied with the work you did to improve your paper. Enclosed please find the reviews of your manuscript.

The reviews are quite favourable and suggest that subject to minor revisions, your paper could be accepted for publication. Please pay special attention to the comments and corrections by Reviewer 1, I find them very useful. As for the comments of the second reviewer, I think your background and reference list is quite complete.

Please consider these suggestions, and I look forward to receiving your revision.

Reviewer 1 ·

Basic reporting

The English of the manuscript has been improved (but see comments for specific changes). In general, the article is relatively well-written with adequate citations, figures, and tables.

Experimental design

This paper provides important ecological and life history data for a species of salamander whose basic biology is essentially unknown. The methods used are appropriate and adequately explained .

Validity of the findings

In general, the data in this manuscript are robust and reasonably analyzed. Some of the conclusions in the new version go beyond what can be stated from the data (see specific comments below) and some of the statistical differences seem like they might be so minor as to be of little biological meaning. While it is impressive how many individual salamanders the authors were able to sample, statistical analysis of such a large data set will nearly always show differences between groups. The question is then whether such differences mean anything. There are a few claims of priority and novelty in the conclusions that are unnecessary and detract from the genuine contribution of the manuscript, which is to provide a large amount of data about a species for which almost noting was previously known.

Additional comments

Specific comments by line number:

Lines 30–31: It is inaccurate to say that this species is one of the most threatened in South America. There are many frogs that have not been seen in decades despite intense search efforts and are likely extinct. This species seems more understudied than highly threatened (even by the authors’ own criteria in the conclusion). Also, change “on the Colombian conservation agenda” to “for the Colombian conservation agenda”.

Lines 33–34: Change “presenting a narrow elevational range only within the Cundinamarca department” to “occurring only in the department of Cundinamarca within a narrow elevational range”

Line 36: change “covered” to “cover”

Line 43: Change “a multivariate analyses” to “Multivariate analyses” (if multiple analyses were conducted) or “A multivariate analysis” (if only one analysis was done).

Line 44 and elsewhere: Change “not detection” to “nondetection”

Line 49: Delete “distribution of this”

Line 54: Spell out genus name when it is the first word in a sentence

Line 78: Change period to comma after “Frost”

Line 89: While this section has been rephrased, it is still incorrect as written. Does the 12% figure refer only to South American species of Bolitoglossa? If so, it could be correct. As written, the sentence seems to refer to 12% of all Bolitoglossa species. The vast majority of Bolitoglossa species descriptions contain information on microhabitat use and habitat preference. If all species with some information on microhabitat use or habitat preference were included, the figure would be far higher than 12%. I agree that we do not know much about many species of Bolitoglossa, but we have at least some information on microhabitat use for most of them.

Line 102: delete “the” before “primary cloud forest”

Lines 98, 105: Snout-vent length should be spelled out and then abbreviated on the first mention, not the second.

Line 108: I recommend reporting SVL to only one decimal place (i.e. 37.6 mm)

Line 115: Change “describe the geographic range extension” to “describe the geographic range”

Line 138 and elsewhere: Change “altitudinal” to “elevational” and “altitude” to “elevation”

Line 162: Change “was recorded” to “were recorded”

Lines 184, 191: “Life forms” usually means something different than this (basically, living things in general). I would change this to “growth forms” or just “categories”.

Lines 198–202: Which of these were dependent variables and which were independent variables?

Line 206: Delete “analysis” (because it’s already included in PCA)

Line 217: Change “Smirnov’s” to “Smirnov”

Line 219: Change “data size” to “sample size”

Line 234: Avoid words like “remarkable” because they imply a value judgment. I would change this to “high”

Line 246: Again, you need to say what you mean by “population structure”. In previous sections of the manuscript, you say “population size-structure”. I think this is still slightly misleading because “population size” usually refers to the number of individuals in a population. I recommend changing this to “population age-structure”, because it seems like the main point is to classify individuals into life stages (adult, juvenile, etc).

Line 262, 263: change “Km” to “km”

Line 267: Change “sighted” to “found”

Line 277: Change “squared coefficients” to “R-squared values” or “squared correlation coefficients”

Line 278: Delete comma after “variables”

Line 282: Delete “analysis” after “PCA” and change “retrieved” to “accounted for”

Line 284: Change “overlapped” to “overlapping”

Line 295: Why is “best” in quotation marks here? This is the right word and it has an understood meaning in this context; there is no need to put it in quotes.

Lines 307, 308: If these are correlation coefficients, they should be “r”, not “R”.

Line 312: Change “being more abundant” to “being higher”

Lines 310–312: I don’t think that much is gained by analyzing the sampling events separately. Conditions vary daily and what happened on any one particular day is not particularly interesting, whereas patterns of activity that are consistent across days are interesting. I recommend combining all sampling occasions and seeing if there are peaks of activity and if they are significantly different. Also, Figure 4 would be easier to read if it were centered on midnight rather than noon (so that the activity period were in the center of the figure).

Line 320: Change “in contrast with” to “in contrast to”

Line 321: Change to “variation in salamander SVL was between peaks was consistent…”

Lines 327–328: High compared to what? Most species of Bolitoglossa have a generation time of two years, and they likely all transition quickly from neonates to juveniles. Given that the sampling locations were spread throughout a large part of the year, this seems about what would be expected.

Line 330: Change “of the individuals” to “of individuals”

Line 332: Change “SVL median value” to “median SVL “

Lines 334: I don’t think that the timing of the recruitment peak can be determined, because no samples are available between October and March.

Line 338: Change “tiny” to “small” or “very small”. There are much smaller salamanders in the Neotropics, so I don’t think that calling it “tiny” is accurate.

Line 337: I must insist: if you are giving diagnoses from other species within the genus, this section cannot be titled “Morphological variability”. This is basically a redescription without reference to the type material. At a minimum, you could say “Morphological variability and comparisons to other species” or something like that, but there is very little data on actual morphological variability presented in the section.

Lines 349–350: No species of South American Bolitoglossa has a complete lack of webbing. I would delete “absent”. Compared to species in Mesoamerica or North America, the least webbed South American Bolitoglossa is highly webbed.

Line 352: Change “for having” to “by” or “by having”

Line 353 and elsewhere: Spell out genus name when it is the first word in a sentence.

Line 353: All of the digits are typically referred to as “toes” or “digits”, not “fingers”. It is also important to specify whether this refers to digits on the hand or foot.

Line 359: What is meant by “digital depressions”? This is not a standard term in salamander morphology.

Line 373: Change “exhibited” to “exhibits”

Line 385 and elsewhere: I do not understand what “pits” refers to in the context of color pattern.

Line 390: This is almost universally referred to as the “mental gland”, not the “submental gland”.

Line 394: Change “added” to “add”

Line 395: Change “have allowed” to “allowed”. I also recommend making this section shorter and just focusing on how examination of this population provided detailed ecological and morphological data that was previously lacking for the species.

Line 398: “Enormous” implies a value judgment and should be deleted. “Efforts” is fine.

Line 434: Italicize “B. pandi”

Line 438: Change “opposite or absent” to “a negative correlation or no correlation”

Line 452: Do not start a sentence with “Also”. You could say “Additionally”

Line 453: Change “that agrees” to “which agrees”

Lines 468, 470, elsewhere: Change “Bolitoglossines” to “bolitoglossines” or “Bolitoglossa”. “Bolitoglossines” is not a recognized taxonomic rank and thus should not be capitalized.

Line 480: I don’t understand what is meant by “suggesting independence of climatic variability”

Line 495: Delete “Meanwhile” or change it to “However”

Line 497: What is meant by use of resources? The only thing measured here was when salamanders were found and where; no resources (aside from perch height, which is essentially an unlimited resource in the forest) was measured. I would focus on timing and not bring resources into it.

Lines 498–501: Avoid statements of priority and novelty; have the authors read every thesis, paper, and report on tropical salamander activity? Let the results speak for themselves. It is clear that we have learned a lot from this study, without saying that it is the first to demonstrate a point.

Lines 501–504: It’s not that the data are insufficient, but rather that these aspects of salamander ecology were simply not part of this study. I recommend deleting this sentence entirely.

Lines 506–507: It’s not clear to me how life history strategy relates to anything presented in this paper. I recommend deleting this sentence.

Lines 510–513: Adaptive radiation has to do with morphological differences between species, not within them. This sentence is inaccurate.

Line 514: Delete “the” before “suitable habitat conditions” and change “predicting” to “predictor”

Line 515: “Across the Andes” makes it sound like this species has a distribution throughout the mountain range. This should be deleted.

Lines 517–520: I don’t understand what is meant by this sentence. Extinction rates refer to species, not populations. If the authors are referring to population extirpation rates, these are not possible to estimate if the species has not been extirpated at known localities. “Predictive scenarios” doesn’t really mean anything; do the authors mean “predictions”, and if so, predictions of what? Honestly, I don’t think that much can be said about the conservation of this species based on the data presented here, except that it is more widely distributed and more abundant than previously known.

Line 537: Change “is” to “are”

Line 539: Change “as” to “such as”. I would delete this entire sentence; ecological studies are much less important than accurate distributional data in nearly all conservation planning. It doesn’t matter when a species is active or what microhabitat it uses if a forest is destroyed, which is the level at which most conservation planning in the Neotropics (and elsewhere) is done.

References: There is a relatively high degree of self-citation in this manuscript. I would avoid this; Zootaxa was just removed from the list of indexed journals for exactly this problem and we don’t want PeerJ to be next.

Figure 2: Are the “R” values correlation coefficients (in which case they should be “r”) or R-squared? If they are correlation coefficients, it is more appropriate to report them as R-squared values. It would also be worth pointing out in the text that although these values may be statistically significant, they are quite small and might not be very biologically significant.

Figure 4: As stated previously, I would combine data across sampling events and to focus on temporal differences.

Figure 5: Looking at these plots, it seems like the results of the nested ANOVA may not be saying much. That test would only say if there were a difference between any two time periods. Overall, there is very broad overlap in abundance and salamander size across all times; I don’t think that much can be concluded from this.

·

Basic reporting

The paper is important for the knowledge of a scarce and poorly known species of Andean bolitoglossine. The manuscript is clear and the structure shows standard sections requested. Figures are relevant. I accept it for publication with minor revisions.

Experimental design

This research is relevant; Methods used show sufficient details.

Validity of the findings

Important data are given to define the distribution and conservation status of B. pandi. It's an important paper for its natural history.

Additional comments

Good paper in general. However, please check well your manuscript with an english speaker.
You can also include in the discussion section an overview (some sentences) of the conservation status of the genus Bolitoglossa and other Salamander species, which are the most endangered vertebrate group on Earth. Be that as it may, more than 41% of amphibian species are thought of as ‘endangered’, including 54.7% of the Urodela group. North and Central America and Asia are most severely affected by the continued destruction of habitats and the spread of chytrid (Bd). See my previous book : Hernandez A. 2016. Etude sur les Urodèles en voie de disparition. Edilivre editions, Paris. 120 p.

---

## Round 0.3 · accepted · Accept

Thank you for taking the time to revise and resubmit your manuscript. I have now read through your paper as well as your letter in response to the reviews. I think that you have successfully addressed all of the concerns raised very well, and would like to accept your manuscript for publication in PeerJ. Congratulations.

Thank you for all the hard work you have put into this. Your paper makes a strong contribution to the literature and I look forward to seeing it published.